# Evaluating the comprehensive water resources utilization level in China: Dynamic distribution analysis and spatial convergence insights

Xiongtian Shi(iD)[1*◉], Chao Li[2◉], Zhengyong Yu[1◉]

**1** School of Business Administration and Tourism Management, Yunnan University, Kunming, China,
**2** Party School of Liaocheng Municipal Committee, Liaocheng, China

◉ These authors contributed equally to this work.
* shixiongtian@stu.ynu.edu.cn

**Data availability statement:** All relevant data are within the paper and its Supporting Information files.

**Funding:** The Yunnan University Research Foundation Program(Grant No. KC-23233830); Youth Project of Humanities and Social Science Foundation of Ministry of Education (22YJC790039). The funders had no role in study design, data collection and analysis,

## Abstract

This research provides an overview of the comprehensive water resources utilization level (CWRULE) in China, highlighting its significance in national water management. The study aims to evaluate performance and trends in CWRULE across various regions. Employing methods such as the Dagum Gini coefficient, spatial kernel density estimation, and spatial convergence models, the analysis explores regional disparities, distribution dynamics, and convergence trends. Key findings indicate that while national water resources management has improved annually, significant disparities persist between the coastal eastern and central regions versus the western and northeastern regions, where CWRULE indicators remain relatively low. Notably, the convergence speed in the central, western, and northeastern regions increases significantly after controlling for variables, showcasing the beneficial impacts of policy support, economic development, and technological advancements. In contrast, the eastern region exhibits weak convergence, underscoring the necessity for targeted strategies to enhance water resources management and efficiency.

## 1. Introduction

With the rapid industrialization and urbanization of China, issues related to water scarcity and water pollution have become significant bottlenecks impeding the country's high-quality economic and social development [1]. Urbanization not only involves the concentration of populations in cities but also entails profound changes in economic, social, cultural, and natural landscapes. The swift increase in urban populations and industries has exacerbated the demand for domestic and industrial water, while the resulting discharge of industrial wastewater poses a serious threat to the efficiency of water resources utilization [2,3]. Nevertheless, improvements in urban water infrastructure and adjustments in industrial structures can alleviate this pressure to some extent. In response to these challenges, the State Council issued the "Opinion on Implementing the Strictest Water Resources Management System" in 2012, followed by the "Action Plan for Water Pollution Prevention and Control" in 2015 [4]. These measures introduced the "Three Red Lines" to control total water use, improve

decision to publish, or preparation of the manuscript.

**Competing interests:** NO authors have competing interests.

water use efficiency, and manage pollutant discharge in functional zones [5,6]. These policies significantly enhanced the national water quality environment and achieved the control targets outlined in the Thirteenth Five-Year Plan by 2020, providing crucial support for China's economic and social development [7].

Globally, the utilization and management of water resources face unprecedented challenges, driven primarily by population growth, economic development, and climate change. For instance, Zhang et al. emphasized that hydrological changes induced by climate change have significantly impacted water availability in arid and semi-arid regions, necessitating adaptive management strategies [8]. In Europe, the widely adopted Integrated Water Resources Management (IWRM) framework has played a crucial role in balancing water use efficiency and ecological sustainability [9]. A recent study demonstrated that coordinated policy implementation in European Union countries reduced regional disparities in water use efficiency by 20% over a decade [10]. Additionally, the adoption of innovative agricultural technologies, such as drip irrigation, in the Middle East has effectively improved water use efficiency by 30% [11]. In developing countries, investments in water treatment facilities and infrastructure have yielded significant results. For example, research on rural water resource utilization in India revealed that excessive reliance on groundwater and the degradation of surface water bodies have led to a substantial reduction in water-covered areas (WCA), alongside severe damage to aquatic ecosystems and vegetation cover [12,13]. Bhuyan and Deka further highlighted that in the Nagaon District of Assam, India, the number and size of wetlands have significantly declined, posing serious threats to the habitats and ecological conditions of flora and fauna dependent on these water bodies [14]. These changes not only present challenges to the sustainable use of water resources but also underscore the critical importance of community involvement in policymaking for ecohydrological conservation and restoration.

Recent empirical studies worldwide highlight the critical importance of sustainable water resource management, particularly in the context of increasing concerns over water scarcity and quality degradation. According to United Nations data, by 2025, over 1.8 billion people globally are expected to face water shortages, with economic losses due to water pollution reaching billions of dollars annually. Research in arid regions has demonstrated that innovative water conservation measures can significantly enhance water use efficiency; for instance, a study found that the adoption of drip irrigation technology improved agricultural water efficiency by 30% [15]. Comparative analyses in Europe have revealed that effective policy frameworks and integrated water resource management can promote spatial convergence among different regions, particularly following the implementation of unified water management policies, where water use efficiency in some areas increased by 20% [16]. Furthermore, studies in developing countries indicate positive outcomes from investments in infrastructure and technology, noting that advanced water treatment facilities can enhance water resource utilization rates by 25% in certain regions [17]. Collectively, these studies stress the need to integrate qualitative and quantitative factors in management, focusing not only on water quantity but also on quality and ecological impacts.

To address water resource management challenges, developing an evaluation system that integrates both water quality and quantity is crucial. Previous studies have focused on three areas: First, evaluations of water use analyzed national, watershed, and provincial water footprints, exploring the status and trends of utilization across regions [18–20]. Second, water quality assessments have received widespread attention, with various studies providing detailed evaluations of river basin water quality [21–23]. Third, coupled assessments of water quality and quantity offer a more comprehensive understanding of utilization by examining stress levels and their relationship with economic activities [24,25]. However, existing research often falls short in comprehensively evaluating both aspects. Most studies concentrate on

usage, with assessments typically focusing on wastewater discharge volumes rather than overall aquatic health [26]. Moreover, there has been a lack of in-depth exploration into regional differences and the spatio-temporal evolution of utilization. These gaps hinder a holistic understanding of comprehensive water resource use.

Based on this, this research incorporates water quality into the Comprehensive Water Resources Utilization Level (CWRULE) evaluation framework and employs water footprint theory to assess China's CWRULE comprehensively. Specifically, it evaluates the comprehensive utilization levels across China's provinces, municipalities, and autonomous regions, considering economic influences. The analysis focuses on regional disparities, their sources, and the patterns of spatio-temporal evolution in utilization. This research holds significant implications not only for China but also for the global community, particularly for other developing countries. Efficient resource use is crucial for achieving sustainable development goals, especially where resources are scarce. Effective management and policies are essential to mitigate pressures—a challenge all developing nations face. By providing a detailed analysis of CWRULE in China, this research aims to enhance global water resource management and promote sustainable use. China's experiences in this domain will offer valuable references for other developing countries as they strive for economic growth while ensuring sustainable water supplies, thereby supporting their high-quality development and contributing to the global pursuit of sustainability.

The innovation of this research lies in integrating water quality into the CWRULE evaluation framework, thus providing a comprehensive perspective that addresses the existing imbalance between quality and quantity in resource management. Furthermore, it explores the CWRULE across various provinces, revealing significant disparities in utilization efficiency between coastal eastern regions and western and northeastern regions. The research addresses the traditional focus on water usage while emphasizing the necessity for targeted management strategies.

The primary objective is to conduct a comprehensive assessment of the CWRULE across China's provinces, municipalities, and autonomous regions, analyzing underlying factors and regional disparities. It also examines the spatio-temporal evolution patterns, providing insights into dynamic changes in resource management. Additionally, through empirical analysis, this research aims to offer scientific support for formulating and implementing water resource management policies, contributing to achieving sustainable development goals and providing valuable experiences and references for global management.

The remainder of this research is organized as follows: Section 2 introduces methods for measuring CWRULE, including variable selection, regional difference analysis, distribution dynamics, and methods for calculating spatial convergence. Section 3 provides a detailed analysis of measurement results, regional differences, and spatial convergence. Section 4 summarizes findings, offers recommendations, and discusses future directions for research in developing countries. To visually present the research framework, a flowchart of empirical steps has been created and is displayed in Fig 1.

## 2. Materials and methods

### 2.1. Measurement model

**2.1.1. CWRULE measurement model.** This research utilizes the comprehensive assessment model for water quality and quantity and the coupling model for water quality and quantity to calculate CWRULE, providing a more comprehensive assessment of water resources utilization.

(1) Assessment model for water quality and quantity

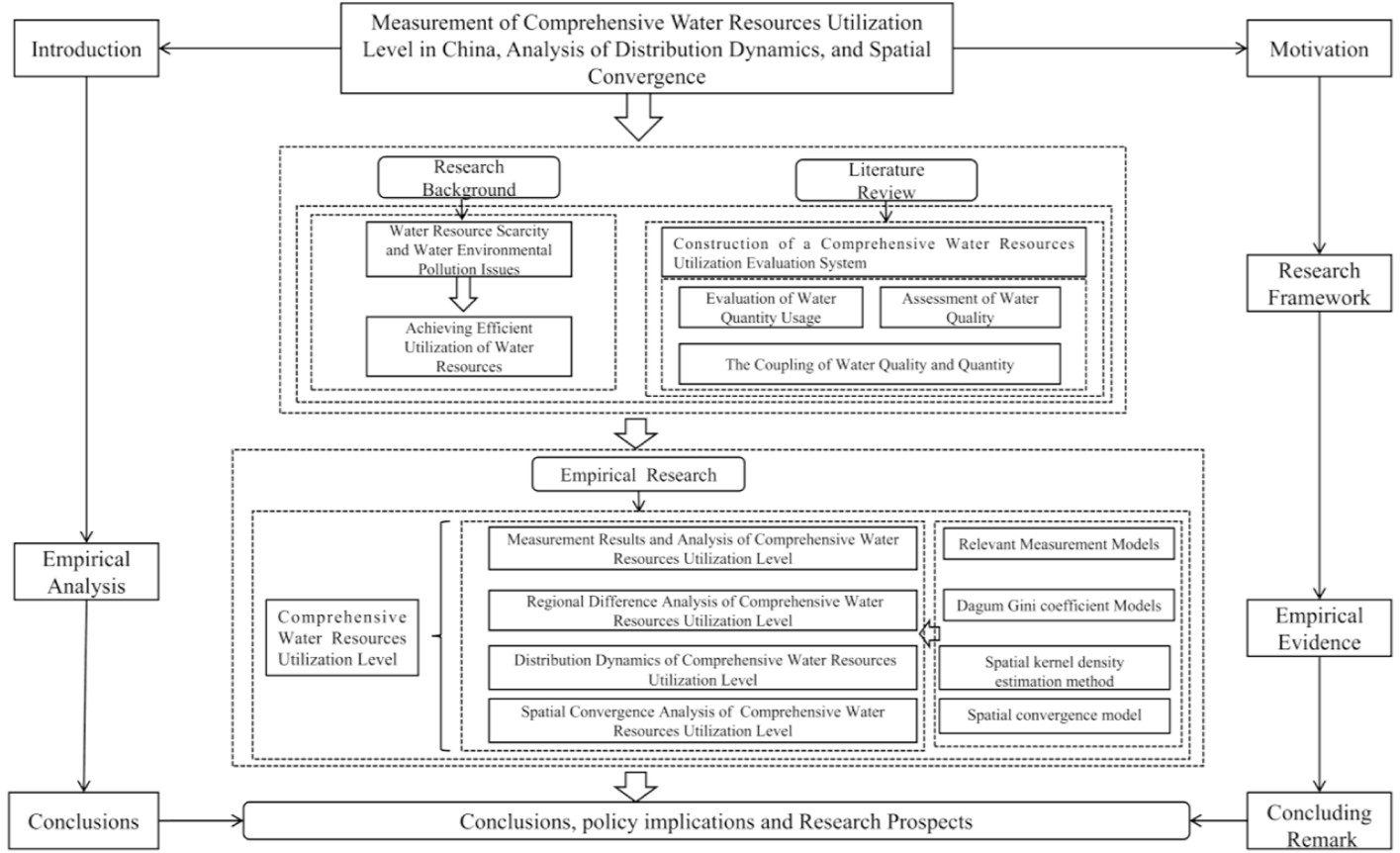

**Fig 1. Flow chart of CWRULE research framework.**

The construction of the comprehensive assessment model for water quality and quantity is as follows:

In the comprehensive assessment of water quality, the volume of wastewater discharge and the content of pollutants within it are used as assessment indicators. $x_{ij}$ is set to represent the volume of wastewater discharge per trillion GDP and other pollutant discharge indicators for region $i$ in year $j$. Since this is a negative indicator, where a larger value indicates poorer water quality, it is necessary to standardize it for analysis. The standardization formula is as follows:

$$z_{ij} = \frac{\max\left(x_j\right) - x_{ij}}{\max\left(x_j\right) - \min\left(x_j\right)} \tag{1}$$

In this formula, $z_{ij}$ is the standardized result of $x_{ij}$, where i = 1,2,3,…,30; and $j$ = 1, 2,3, …,11.

$$z_j = \sqrt[3]{z_{1j} \times z_{2j} \times z_{3j}} \tag{2}$$

The $z_{ij}$ represents the water quality index for a specific region in year $j$. A higher index indicates more effective water pollution control and better water quality in that region.

(2)Assessment model for water consumption assessment model

The construction of the comprehensive assessment model for the coupling model for water quality and quantity is as follows:

Water consumption is commonly measured by the water footprint. Referencing the 'Water Footprint Assessment Manual' by Hoekstra et al. [27], along with research by related scholars [23,28], this paper first calculates the water footprint, then divides it by the regional GDP total, and standardizes the result to determine the level of water resources consumption. The formula for calculating the water footprint is

$$TWF = IWF + EWF \qquad (3)$$

$$IWF = AWU + IWU + UDWU + EWU - VWIE \qquad (4)$$

$$EWF = ITV / GDP \times T\,W \qquad (5)$$

$$VWIE = OTV / GDP \times T\,W \qquad (6)$$

TWF represents the total water footprint, a comprehensive measure encompassing all water used within a specified region. IWF, the internal water footprint, accounts for water use internally within the region including agricultural (AWU), industrial (IWU), urban domestic (UDWU), and ecological (EWU) water uses, offset by the virtual water volume of local exports (VWIE). EWF denotes the external water footprint, calculated as the import trade value (ITV) divided by the Gross Domestic Product (GDP), multiplied by the total water use (TW) in the region. The calculation method for the Virtual Water Volume of Local Exports (VWIE) is by dividing the Export Trade Value (OTV) by the GDP, and then multiplying by TW. This framework highlights both the direct and indirect water usage and is crucial for assessing the sustainability of water resources management in the area.

TWF is a negative indicator, and its normalization formula is:

$$TWF_j = \frac{\max\left(TWF_j\right) - TWF_{ij}}{\max\left(TWF_j\right) - \min\left(TWF_{ij}\right)}. \qquad (7)$$

(3)Assessment model for the coupling model for water quality and quantity
The water quality model and the water footprint model have been integrated to form a comprehensive model for the utilization level of water resources.

$$CWRULE = z_j \times TWF_j \qquad (8)$$

A higher value of CWRULE indicates a greater efficiency in the comprehensive utilization of water resources.

**2.1.2. Dagum Gini coefficient and its decomposition.** The Dagum Gini coefficient is applied to analyze the regional variability of CWRULE in China, making it particularly suitable for multi-regional and multi-level comparative analysis. Compared to traditional Gini coefficients, the Dagum Gini coefficient can effectively decompose the overall inequality, identifying intra-regional (within-group) and inter-regional (between-group) differences. By decomposing the Gini coefficient, this study can clearly pinpoint disparities in resource utilization across different regions and groups, along with their underlying sources. This decomposition method provides profound insights for understanding changes in CWRULE and their potential policy implications. Referring to related scholars [29,30], the Dagum Gini coefficient and its decomposition are calculated using the following:

$$\begin{cases} G = \dfrac{\sum_{i=1}^{k}\sum_{m=1}^{k}\sum_{j=1}^{n_i}\sum_{r=1}^{n_m}\left|y_{ij}-y_{mr}\right|}{2n^2\mu} \\[4mm] G_{ii} = \dfrac{\sum_{j=1}^{n_i}\sum_{r=1}^{ni}\left|y_{ij}-y_{ir}\right|}{2n_i^2\mu_i} \\[4mm] G_{im} = \dfrac{\sum_{j=1}^{n_i}\sum_{r=1}^{n_m}\left|y_{ij}-y_{mr}\right|}{n_in_m(\mu_i+\mu_m)},\mu_m\leqslant\cdots\leqslant\mu_i\leqslant\cdots\leqslant\mu_k \\[4mm] G = G_w+G_{nb}+G_l,G_{gb}=G_{nb}+G_l,G_w=\sum_{i=1}^{k}G_{ii}p_is_i \\[2mm] G_{nb} = \sum_{i=2}^{k}\sum_{m=1}^{i-1}G_{im}(p_is_m+p_ms_i)D_{im} \\[2mm] G_l = \sum_{i=2}^{k}\sum_{m=1}^{i-1}G_{im}(p_is_m+p_ms_i)(1-D_{im}) \\[2mm] D_{im} = \dfrac{d_{im}-p_{im}}{d_{im}+p_{im}} \end{cases} \tag{9}$$

where $G$ denotes the overall Gini coefficient of CWRULE, and $y_{ij}$ denotes CWRULE in the $j$-th province of the $i$-th region, $k$ is the number of regions, $n$ is the number of provinces, and $\mu$ is the average value of CWRULE in each region, and $G_{ii}$ is the Gini coefficient of the $i$-th region, and $G_{im}$ is the Gini coefficient between the $i$-th and $m$-th regions, and $D_{im}$ is the relative impact of CWRULE for cultivation between the $i$-th and $m$-th region. $d_{im}$ is the difference in CWRULE between regions, and $p_{im}$ is the difference in CWRULE between the i and m regions, $y_{mr}-y_{ij}$ is the mathematical expectation of the sum of the $>0$ sample values in the $i$ and $m$ regions.

**2.1.3. Spatial kernel density estimation method.** Spatial kernel density estimation is a powerful non-parametric statistical method used to explore the distribution patterns of spatial data. By applying this method, the research can visually identify the concentration levels and distribution characteristics of CWRULE across different regions. Spatial kernel density estimation reveals hotspots and cold spots of resource utilization within specific areas, offering targeted data support for policymakers to help optimize water resource management and allocation strategies.

Spatial autocorrelation testing is an essential initial step in examining the dynamics of spatial kernel density distributions [31]. This process is crucial for verifying the precision and pertinence of the analysis, and it facilitates a logical interpretation of the spatial traits of CWRULE. By employing the Moran's I statistic [32], the spatial correlation of the CWRULE distribution among Chinese provinces can be evaluated, establishing a robust basis for more detailed dynamic spatial distribution studies. The formula for calculating Moran's I is presented as follows:

$$\text{Moran's I} = \frac{\sum_{i=1}^{n}\sum_{j=1}^{n}W_{ij}\left(Y_i-\overline{Y}\right)\left(Y_j-\overline{Y}\right)}{S^2\sum_{i=1}^{n}\sum_{j=1}^{n}W_{ij}} \tag{10}$$

$$S^2 = \frac{1}{n}\sum_{i=1}^{n}\left(Y_i-\overline{Y}\right),\overline{Y}=\frac{1}{n}\sum_{i=1}^{n}Y_i \tag{11}$$

Where $Y_i$ and $Y_j$ represent the observed CWRULE values for provinces $i$ and $j$ respectively, and $W_{ij}$ is the spatial adjacency weight matrix, implemented as a 0-1 matrix.

The spatial kernel density estimation approach is employed to study the distribution dynamics of CWRULE across China. This method enhances the conventional kernel density estimation, depicted in equations (12) and (13), by integrating temporal and spatial variables. This refined

method utilizes continuous density curves to illustrate the distribution states of the random variables during spatio-temporal development, as demonstrated in equations (14) and (15).

$$f(x) = \frac{1}{Nh}\sum_{i=1}^{N}K\left(\frac{X_i - x}{h}\right) \tag{12}$$

$$K(x) = \frac{1}{\sqrt{2\pi}}\exp\left(-\frac{x^2}{2}\right) \tag{13}$$

$$f(x,y) = \frac{1}{Nh_xh_y}\sum_{i=1}^{N}K_x\left(\frac{X_i - x}{h_x}\right)K_y\left(\frac{Y_i - y}{h_y}\right) \tag{14}$$

$$g(y\,|\,x) = \frac{f(x,y)}{f(x)} \tag{15}$$

Where $f(x)$ denotes the random variable $x$ density function; $N$ is the number of observations; and $h$ denotes the bandwidth; $K(x)$ denotes the random variable $x$ the kernel function of the random variable; $f(x,y)$ denotes the joint density function of $x$ and $y$. $g(y\,|\,x)$ denotes the distributional state of y under the x condition.

**2.1.4. Spatial convergence model.** The spatial convergence model is used to analyze the convergence trends of different regions concerning CWRULE, specifically whether resource utilization levels in various areas tend to align. This model takes spatial autocorrelation into account, effectively capturing the dynamic interactions between regions. By employing this model, the research can delve into the impact of regional development policies on the efficiency of water resource utilization, revealing the effectiveness and challenges of policy implementation in different areas.

This research employs absolute β convergence, conditional β convergence to examine the evolutionary trends of CWRULE across different regions in China. Drawing on the research on convergence by Barro R. J [33].and others [34,35], absolute β convergence suggests that under identical structures, CWRULE will converge to the same level over time, with the growth rate of indices in regions with lower levels of CWRULE being faster than those in higher-level regions. Conditional β convergence, posits that the growth rate of CWRULE depends not only on the initial development level but also on other factors, leading regions to converge towards their own steady states. Utilizing a spatial adjacency matrix (0-1 matrix), this research introduces the Spatial Durbin Model (SDM) for spatial β convergence analysis, following the methodologies of related scholars [36,37]. The formula for absolute β convergence in this research is expressed as follows:

$$\ln\left(\frac{Y_{i,t}}{Y_{i,t-1}}\right) = \alpha + \beta\ln Y_{i,t-1} + \rho\sum_{j\neq i}^{n}W_{i,j}\ln\left(\frac{Y_{i,t}}{Y_{i,t-1}}\right)$$
$$+ \theta\sum_{j\neq i}^{n}W_{i,j}\ln Y_{i,t-1} + +\varphi_i + \pi_t + \mu_{it} \tag{16}$$

The formula for conditional β convergence is expressed as:

$$\ln\left(\frac{Y_{i,t}}{Y_{i,t-1}}\right) = \alpha + \beta\ln Y_{i,t-1} + \rho\sum_{j\neq i}^{n}W_{i,j}\ln\left(\frac{Y_{i,t}}{Y_{i,t-1}}\right)$$
$$+ \theta\sum_{j\neq i}^{n}W_{i,j}\ln Y_{i,t-1} + \gamma lnx_{it} + \psi\sum_{j\neq i}^{n}W_{i,j}lnx_{jt} + \varphi_i + \pi_t + \mu_{it} \tag{17}$$

Where $x_{it}$ represents the control variables. In this research, the selected control variables include per capita GDP, urbanization rate (the proportion of urban population), human capital (the ratio of enrolled students in higher education to the total population), degree of openness (the ratio of goods and services trade value to GDP), and the level of government intervention (the ratio of local government general budget expenditure to GDP).

In summary, the selection of the Dagum Gini coefficient, spatial kernel density estimation, and spatial convergence model provides this research with rich analytical perspectives, helping to uncover regional differences and trends in China's water resource utilization levels. The combination of these methods not only enhances the depth and breadth of the study but also offers solid data support for subsequent policy recommendations.

**2.2. Data specification.** This research focuses on the 30 provinces of mainland China (excluding Hong Kong, Macau, Taiwan, and Tibet) and utilizes provincial panel data from 2011 to 2021 to evaluate the comprehensive utilization level of water resources from a coupled water quality and quantity perspective, as well as to investigate sources of regional differences and spatial convergence. Data is sourced from the China Statistical Yearbook, which provides economic indicators such as GDP and trade values; the China Environmental Statistical Yearbook, offering water quality metrics like wastewater discharge volumes; and the EPS DATA database, which supplements environmental statistics. For any missing data points, interpolation methods have been applied to ensure dataset completeness. However, potential limitations include discrepancies in data accuracy due to varying reporting practices among provinces, which may lead to underreported water usage or pollution levels. Additionally, the temporal coverage might be inconsistent, as some years could have more comprehensive data than others. Geographic diversity among provinces can also affect the generalizability of findings, while a quantitative focus may overlook qualitative aspects of water management. Furthermore, there may be a temporal lag in reflecting the effects of policy interventions in the data, and reliance on external comparisons may introduce biases. By addressing these aspects, the research aims to enhance the validity and reliability of its findings, providing a nuanced understanding of water resource utilization in China.

This research abstracts from real-world entities related to human or animal subjects while ensuring compliance with ethical standards set by institutional or national research committees, as well as internationally recognized guidelines such as the 1964 Helsinki Declaration and its subsequent amendments.

## 3. Empirical analysis

### 3.1. Measurement results and analysis of CWRULE

The measurement results of CWRULE, including key metrics and their variations, are detailed in Table 1, providing a comprehensive overview of the observed trends and patterns. Provinces in the central region, such as Anhui, Henan, Hubei, Hunan, Jiangxi, and Shanxi, show a year-on-year increase in CWRULE values, indicating an improvement in the comprehensive utilization level of water resources, particularly in Shanxi Province, which has the highest average CWRULE value, demonstrating effective water resources management. This trend is consistent with findings from other studies. For example, Rani (2022) reported that targeted regional policies and infrastructure investment significantly enhanced water management efficiency in certain regions in India [38]. In contrast, the western region exhibits significant variability, with some provinces like Gansu and Ningxia having relatively low CWRULE values, indicating lower water resource utilization efficiency. This phenomenon is similar to that observed in Brazil, where regions with less developed infrastructure and policy support also show lower resource management performance [39]. Conversely, Shaanxi and Chongqing

**Table 1. Measurement results of CWRULE.**

| Region | Province | 2011 | 2012 | 2013 | 2014 | 2015 | 2016 | 2017 | 2018 | 2019 | 2020 | 2021 | Mean |
|---|---|---|---|---|---|---|---|---|---|---|---|---|---|
| Central | Anhui | 0.642 | 0.637 | 0.637 | 0.658 | 0.640 | 0.666 | 0.681 | 0.681 | 0.686 | 0.708 | 0.706 | 0.668 |
| | Henan | 0.554 | 0.558 | 0.561 | 0.593 | 0.601 | 0.689 | 0.718 | 0.707 | 0.691 | 0.725 | 0.733 | 0.648 |
| | Hubei | 0.619 | 0.629 | 0.639 | 0.646 | 0.639 | 0.695 | 0.693 | 0.685 | 0.676 | 0.712 | 0.650 | 0.662 |
| | Hunan | 0.597 | 0.598 | 0.599 | 0.607 | 0.615 | 0.658 | 0.672 | 0.665 | 0.659 | 0.699 | 0.676 | 0.640 |
| | Jiangxi | 0.649 | 0.668 | 0.648 | 0.659 | 0.663 | 0.663 | 0.702 | 0.709 | 0.706 | 0.728 | 0.721 | 0.683 |
| | Shanxi | 0.672 | 0.672 | 0.673 | 0.682 | 0.705 | 0.737 | 0.751 | 0.742 | 0.730 | 0.776 | 0.765 | 0.719 |
| West | Gansu | 0.426 | 0.428 | 0.430 | 0.428 | 0.430 | 0.468 | 0.478 | 0.481 | 0.582 | 0.589 | 0.586 | 0.484 |
| | Guangxi | 0.630 | 0.624 | 0.633 | 0.639 | 0.663 | 0.709 | 0.716 | 0.708 | 0.708 | 0.734 | 0.734 | 0.682 |
| | Guizhou | 0.595 | 0.595 | 0.606 | 0.607 | 0.621 | 0.655 | 0.647 | 0.642 | 0.649 | 0.674 | 0.658 | 0.632 |
| | Inner Mongolia | 0.629 | 0.633 | 0.647 | 0.642 | 0.642 | 0.695 | 0.708 | 0.702 | 0.680 | 0.711 | 0.693 | 0.671 |
| | Ningxia | 0.580 | 0.587 | 0.585 | 0.587 | 0.392 | 0.408 | 0.510 | 0.408 | 0.407 | 0.412 | 0.411 | 0.481 |
| | Qinghai | 0.518 | 0.620 | 0.620 | 0.722 | 0.613 | 0.612 | 0.618 | 0.626 | 0.620 | 0.621 | 0.620 | 0.619 |
| | Shaanxi | 0.796 | 0.806 | 0.812 | 0.810 | 0.815 | 0.856 | 0.851 | 0.858 | 0.853 | 0.870 | 0.861 | 0.835 |
| | Sichuan | 0.644 | 0.641 | 0.649 | 0.654 | 0.647 | 0.689 | 0.696 | 0.702 | 0.699 | 0.728 | 0.720 | 0.679 |
| | Xinjiang | 0.353 | 0.184 | 0.169 | 0.186 | 0.242 | 0.289 | 0.322 | 0.354 | 0.279 | 0.336 | 0.317 | 0.276 |
| | Yunnan | 0.486 | 0.495 | 0.494 | 0.503 | 0.506 | 0.508 | 0.544 | 0.540 | 0.538 | 0.550 | 0.544 | 0.519 |
| | Chongqing | 0.843 | 0.847 | 0.847 | 0.850 | 0.856 | 0.901 | 0.907 | 0.907 | 0.908 | 0.917 | 0.918 | 0.882 |
| East | Beijing | 0.985 | 0.983 | 0.982 | 0.981 | 0.976 | 0.975 | 0.976 | 0.979 | 0.976 | 0.974 | 0.976 | 0.979 |
| | Fujian | 0.650 | 0.701 | 0.697 | 0.709 | 0.720 | 0.750 | 0.763 | 0.715 | 0.744 | 0.763 | 0.767 | 0.725 |
| | Guangdong | 0.682 | 0.701 | 0.721 | 0.714 | 0.707 | 0.766 | 0.783 | 0.823 | 0.831 | 0.852 | 0.851 | 0.766 |
| | Hainan | 0.264 | 0.262 | 0.263 | 0.260 | 0.260 | 0.267 | 0.263 | 0.365 | 0.463 | 0.565 | 0.667 | 0.354 |
| | Hebei | 0.449 | 0.454 | 0.491 | 0.515 | 0.586 | 0.656 | 0.690 | 0.709 | 0.709 | 0.719 | 0.694 | 0.606 |
| | Jiangsu | 0.796 | 0.898 | 0.874 | 0.842 | 0.880 | 0.794 | 0.875 | 0.882 | 0.803 | 0.840 | 0.838 | 0.847 |
| | Shandong | 0.435 | 0.459 | 0.480 | 0.490 | 0.498 | 0.568 | 0.612 | 0.607 | 0.588 | 0.655 | 0.694 | 0.553 |
| | Shanghai | 0.854 | 0.862 | 0.862 | 0.879 | 0.885 | 0.897 | 0.908 | 0.912 | 0.909 | 0.914 | 0.911 | 0.890 |
| | Tianjin | 0.941 | 0.942 | 0.944 | 0.944 | 0.947 | 0.961 | 0.961 | 0.960 | 0.959 | 0.963 | 0.962 | 0.953 |
| | Zhejiang | 0.713 | 0.721 | 0.731 | 0.743 | 0.754 | 0.809 | 0.820 | 0.831 | 0.831 | 0.859 | 0.852 | 0.788 |
| Northeast | Heilongjiang | 0.335 | 0.331 | 0.333 | 0.333 | 0.346 | 0.365 | 0.371 | 0.388 | 0.423 | 0.421 | 0.411 | 0.369 |
| | Jilin | 0.428 | 0.429 | 0.431 | 0.431 | 0.531 | 0.561 | 0.466 | 0.475 | 0.480 | 0.585 | 0.589 | 0.491 |
| | Liaoning | 0.615 | 0.631 | 0.645 | 0.632 | 0.631 | 0.755 | 0.764 | 0.779 | 0.778 | 0.783 | 0.786 | 0.709 |

have higher CWRULE values, likely due to superior water resource management policies or enhanced regional economic development, demonstrating a strong correlation between regional economic development and water management efficiency.

Coastal provinces, such as Beijing, Shanghai, Jiangsu, and Zhejiang, generally have higher CWRULE values, benefiting from more developed economies and advanced water management measures. This finding aligns with the results of studies in Europe, where well-established governance frameworks and technological advancements significantly improve resource management efficiency in developed regions [16]. In contrast, the northeastern region, including Heilongjiang, Jilin, and Liaoning, generally shows lower CWRULE values, which could be related to the region's heavy industrial structure and less developed water resources management. This situation is similar to that observed in South Africa, where regions with a heavy industrial base struggle with water management efficiency due to the complex interplay between industrial demands and resource conservation [17].

CWRULE levels and national averages across regions are shown in Fig 2. The national average CWRULE value exhibits a yearly increasing trend, indicating an improvement in

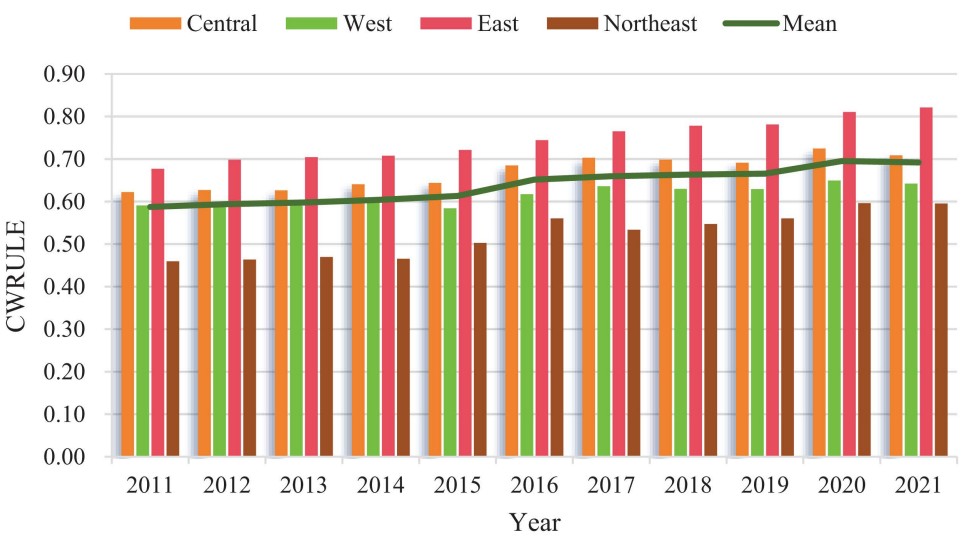

**Fig 2. The CWRULE and mean level in various regions of China.**

national water resources utilization efficiency. Post-2016, the average level stabilizes, suggesting that significant advancements or maturity in water resource management policies and technologies have been reached. The central region consistently maintains high CWRULE values, close to or surpassing the national average, indicating effective water resources management, peaking in 2020, which shows sustained improvement over the decade. The western region has the lowest CWRULE values among all regions, particularly in earlier years, reflecting the challenges posed by geographical and climatic conditions on water resources management. This finding is consistent with studies in arid regions globally, where geographical and climatic constraints pose significant challenges to effective resource management, as observed in studies from the Middle East and North Africa [1].

The eastern region typically has CWRULE values above the national average, demonstrating efficient water resources utilization, benefiting from more developed infrastructure and advanced water management technologies. These results are in line with the findings of other studies, which suggest that developed regions tend to have higher resource management efficiency due to better economic conditions and technological foundations [23]. The CWRULE values in the northeast region, although consistently below the national average, gradually improve, facing specific challenges related to industrial structures or climate. These results are consistent with global cross-national studies on water resource efficiency, which highlight that regions with complex industrial structures and severe climatic conditions tend to have lower efficiency in resource utilization unless supported by strong policy interventions [19].

In summary, the sustained rise in CWRULE values over the years indicates that national investments in water management technologies and policies may have had a positive impact, and the variability in regional performance emphasizes the importance of devising specific water management strategies for each region. The higher CWRULE values in the central and eastern regions could serve as models for other areas. The relatively low performance of the western and northeastern regions indicates a need for targeted policy interventions to address regional issues such as water scarcity in the west and industrial pollution in the northeast. Similar to findings in India and Brazil, region-specific policy adaptations and investments are crucial for effectively addressing these disparities and achieving balanced regional development in water resource management.

## 3.2. Regional difference analysis of CWRULE

The Gini coefficient for CWRULE shows a clear upward trend during the study period, reflecting an increase in inequality, particularly driven by disparities in regional resource allocation (Fig 3).The overall Gini coefficient curve from 2011 to 2021, as illustrated, exhibits a downward trend, indicating that disparities in CWRULE are gradually decreasing nationwide. The intra-group Gini coefficient curve is relatively stable, showing minor fluctuations, which suggests that the level of disparity within groups is consistently stable. The inter-group Gini coefficient also trends downward, indicating that differences in CWRULE between different regions are narrowing. The hyper-variable density Gini coefficient fluctuates significantly, highlighting the impact of extreme values on the overall distribution.

The intra-group contribution rate, as indicated, dominates, suggesting that the primary source of CWRULE disparity arises from within-group differences. The inter-group contribution rate and the hyper-variable density contribution rate are relatively lower, yet they still exert a considerable impact.

Significant fluctuations in the intra-group Gini coefficients across regions are reflected, particularly in the eastern and central regions, where these fluctuations are especially pronounced. This is associated with uneven economic development and differences in resource allocation in these areas. Fluctuations are relatively minor in the western and northeastern regions, indicating a more stable degree of disparity in CWRULE within these areas.

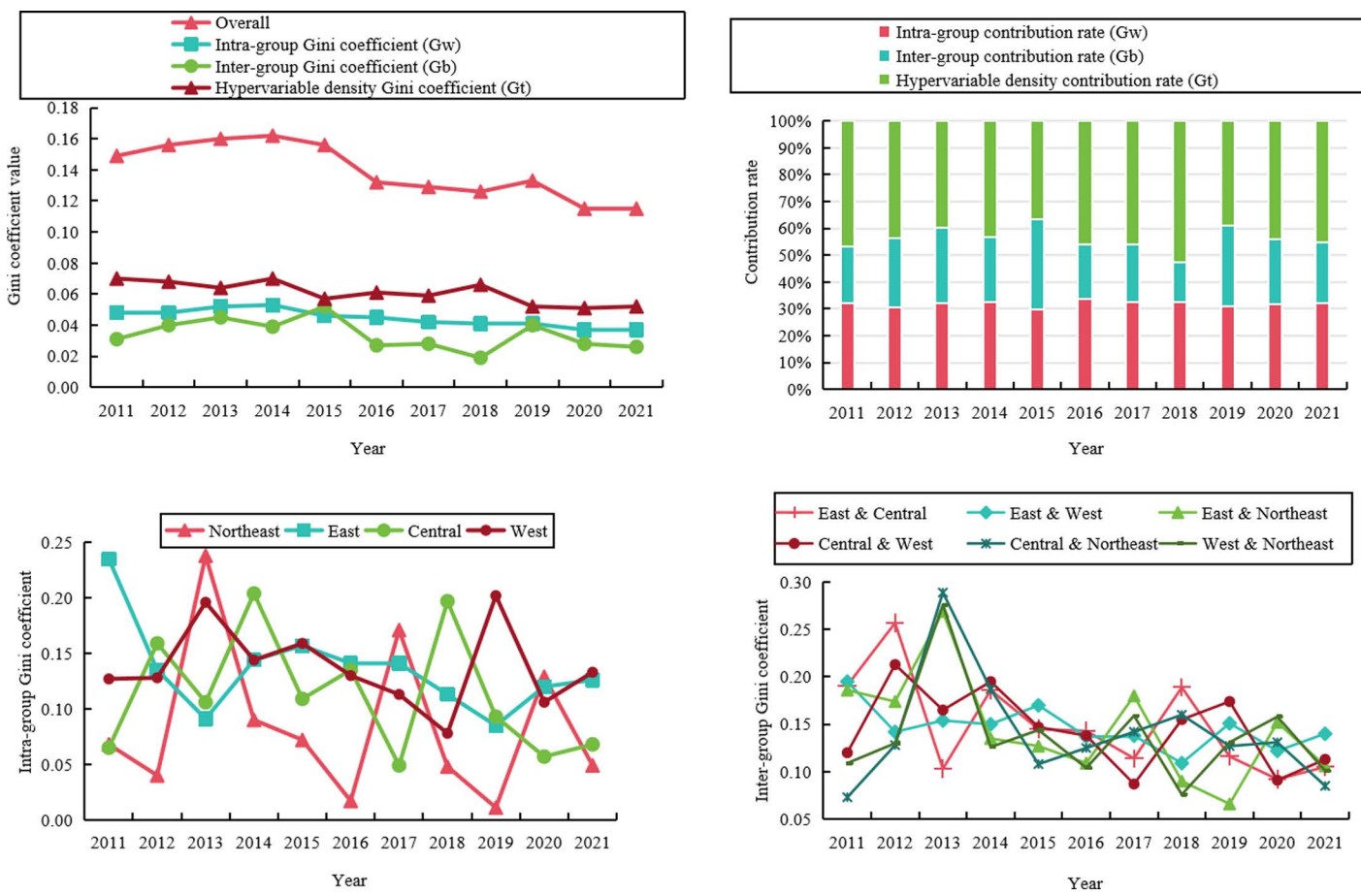

**Fig 3. Changes in the Gini coefficient of CWRULE.**

Changes in the inter-group Gini coefficient are presented. Notable fluctuations in the Gini coefficients between different regional combinations, especially between the east and the west, as well as the east and the northeast, reflect significant differences in economic development levels and water resource management policies among these regions. The Gini coefficients between the central and eastern regions and between the central and northeastern regions are lower, indicating smaller differences in CWRULE between these areas.

Although the overall national level of CWRULE disparity is decreasing annually, significant differences still exist both between and within regions. This requires policymakers not only to focus on national-level policies and measures but also to devise more targeted strategies to reduce disparities in CWRULE, particularly in economically underdeveloped western and northeastern areas. Additionally, enhancing the efficiency and fairness of resource allocation, especially in regions rich in water resources but economically underdeveloped, will be crucial in improving national water resources management levels.

### 3.3. Distribution dynamics of CWRULE

From 2011 to 2021, the Moran's I index values remained consistently above zero and passed the 5% significance level test, as evidenced by the results of the global spatial autocorrelation test (Table 2) and the local Moran's I indices (Fig 4). This indicates significant negative spatial autocorrelation among the provinces in China with respect to CWRULE. Therefore, it is necessary to further analyze the distribution dynamics of CWRULE using the spatial kernel density method.

Using a non-parametric traditional kernel density estimation method, the results reveal the distributional characteristics of CWRULE over time and across regions (Fig 5). The overall kernel density estimates for CWRULE from 2010 to 2021 illustrate a progressive concentration towards higher values, indicating an annual improvement in the comprehensive utilization of water resources at the national level. Regionally, the kernel density plot for the eastern region shows peaks becoming increasingly sharp, suggesting that the distribution of CWRULE

**Table 2. Global spatial autocorrelation measurement results for CWRULE.**

| Index | 2011 | 2012 | 2013 | 2014 | 2015 | 2016 | 2017 | 2018 | 2019 | 2020 | 2021 |
|---|---|---|---|---|---|---|---|---|---|---|---|
| I | -0.095 | -0.144 | -0.134 | -0.147 | -0.133 | -0.114 | -0.128 | -0.102 | -0.098 | -0.13 | -0.160 |
| z(I) | -0.497 | -0.903 | -0.836 | -0.938 | -0.836 | -0.672 | -0.795 | -0.573 | -0.566 | -0.86 | -1.082 |
| p | 0.062 | 0.036 | 0.049 | 0.034 | 0.041 | 0.050 | 0.043 | 0.056 | 0.057 | 0.041 | 0.027 |

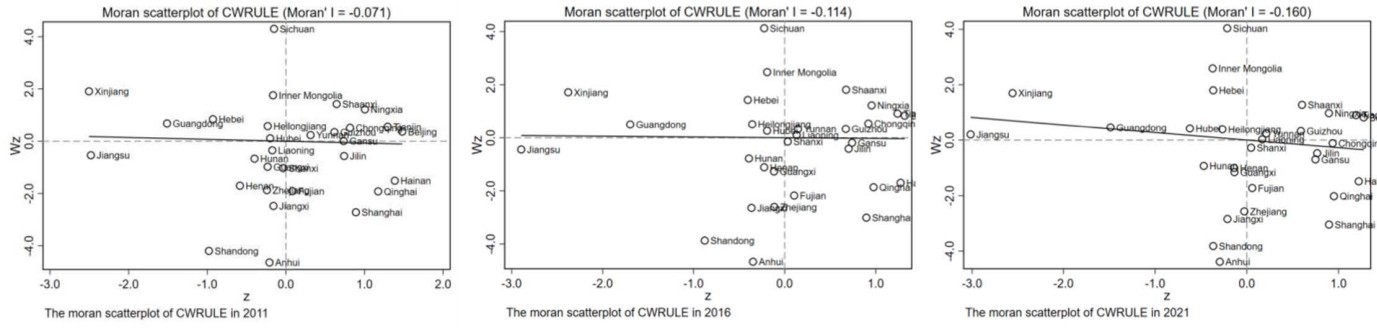

**Fig 4. Local Moran's indices for 2011, 2016, and 2021.**

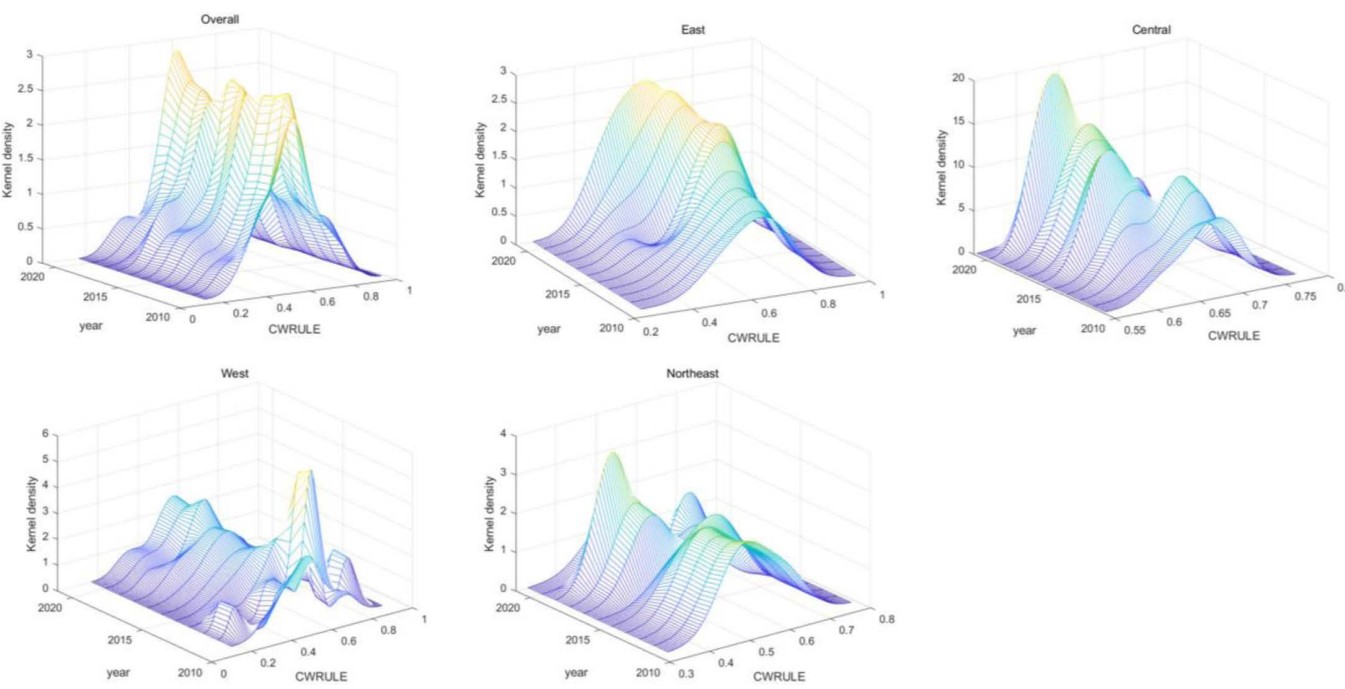

**Fig 5. The traditional kernel density estimation results of CWRULE for various regions.**

is becoming more concentrated at higher levels, indicative of enhanced and more consistent water resources utilization efficiency. In the central region, the kernel density plot exhibits a very sharp peak around 2021, denoting a high concentration of water resources utilization levels and significant improvement in management efficiency during this period. Conversely, the western region's kernel density plot is more dispersed with less pronounced peaks, reflecting a more varied distribution of CWRULE and greater disparities in water resources utilization levels. The northeastern region's plot displays multiple peaks, indicative of a multimodal distribution of CWRULE, suggesting potential significant differences in water resources utilization efficiency among provinces.

There are significant regional disparities in the levels of comprehensive utilization of water resources, and these disparities have evolved over time. While the eastern and central regions have shown improvements in water resources utilization efficiency, the western and northeastern regions, despite some improvements, have experienced relatively slower progress and efficiency gains. This may be associated with factors such as regional economic development levels, water resources management policies, and their implementation efficiency. These analytical findings underscore the need for continued emphasis and investment in water resources management, particularly in the western and northeastern regions. Additionally, there should be an effort to promote inter-regional coordination and resource sharing to enhance the overall efficiency and equity of national water resources management.

The spatial static kernel density and contour maps of CWRULE provide insights into the regional distribution and variations of water resource utilization efficiency (Fig 6).

From spatial static kernel density analysis, the three-dimensional display of the kernel density plot clearly illustrates the relationship between CWRULE values in a given year (Y-axis) and those in adjacent provinces (X-axis), with the kernel density values represented on the Z-axis. Multiple peaks in the plot indicate a concentrated tendency of CWRULE across

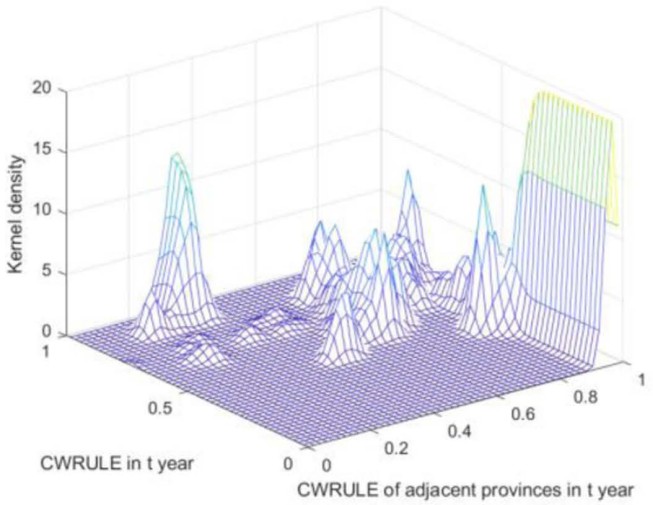 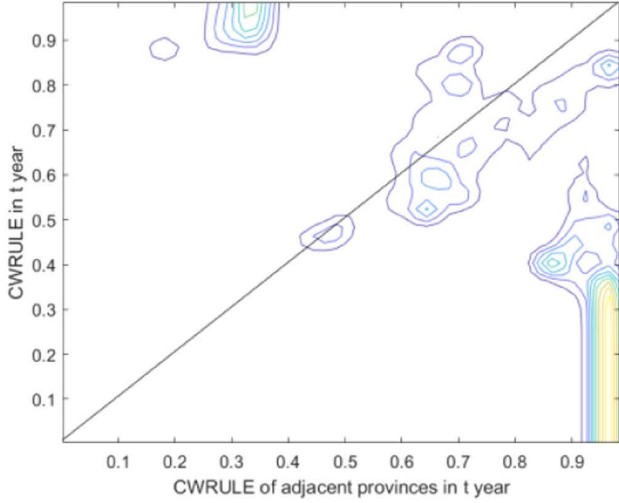

**Fig 6. The spatial static kernel density and contour lines of the kernel density for CWRULE.**

different provinces. The location and height of these peaks suggest a non-uniform distribution of CWRULE levels across regions. The presence of these peaks reflects a clustering of certain provinces or regions, indicating similarities in water resources management and utilization. Furthermore, the multimodality indicated by several peaks near specific values may relate to regional differences in economic development levels, water resources policies, or natural conditions.

From contour line analysis, the contour map depicts the static changes in CWRULE within a particular year. Areas where the contour lines are densely packed indicate regions where changes in CWRULE are concentrated. The diagonal line across the map illustrates the hypothetical scenario where provinces maintain constant CWRULE values over time. The distribution of data points around this line allows for the observation of deviations from this ideal state. Notably, significant deviations, especially in areas with higher CWRULE values, suggest that some provinces experience more pronounced changes in CWRULE compared to others, reflecting variations in the implementation of water resources management policies or the response to water resources challenges.

The spatial distribution and temporal dynamics of CWRULE among different provinces in China reveal significant intra-regional and inter-regional disparities in water resources management and utilization efficiency (Fig 7). The variability and multimodality in the plots underscore that, despite ongoing policy and resource allocations, imbalances in the comprehensive utilization levels of water resources persist across different regions. These analytical findings provide valuable insights for policymakers, aiding in a better understanding of regional disparities and in formulating more effective regional-specific or targeted water resources management strategies.

The spatial dynamic kernel density estimation and kernel density contour lines provide a detailed view of the regional and national trends in CWRULE changes (Fig 7).

The overall plot reveals the trends and density distributions of CWRULE across the nation. Areas where contour lines are densely packed indicate significant concentration of specific CWRULE values, suggesting a high degree of uniformity in CWRULE levels nationally. The diagonal line (ideal line) suggests that if the CWRULE values remain constant over time across provinces, their data points would align along this line. Clearly, most data points cluster

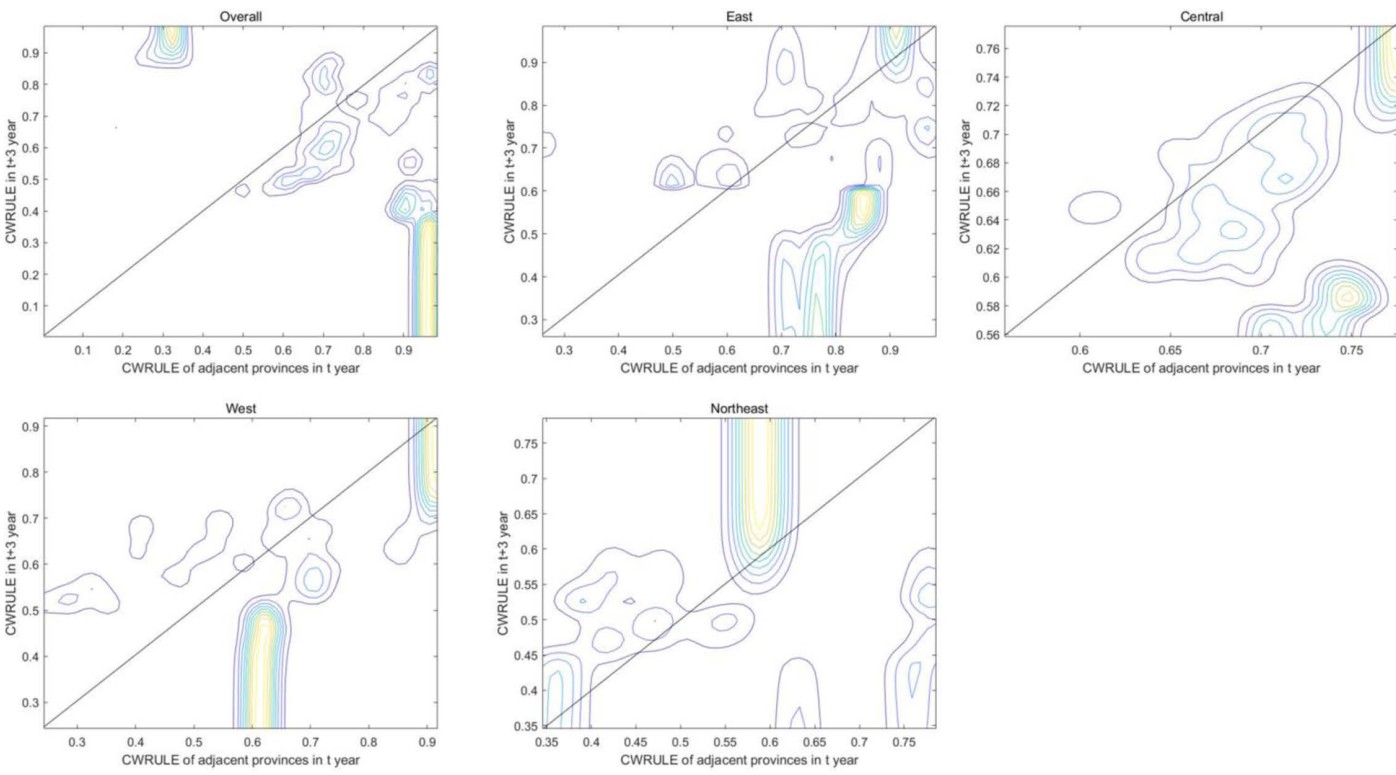

**Fig 7. The spatial dynamic kernel density estimation and kernel density contour lines of CWRULE.**

around this line, indicating minimal change in CWRULE across the majority of provinces. However, significant deviations from this line reflect substantial changes or effects in water resources management strategies in some regions.

The graphical representation for the east shows a more dispersed distribution of CWRULE, with wide-ranging contour spans, indicating complex and varied changes in CWRULE. The fewer concentration areas may suggest significant differences in water resources management and utilization among provinces in the East. The contour lines in the central region are tightly packed and centered in higher value zones, suggesting high consistency and stability in CWRULE, indicative of uniform or effective water resources policies in this area. The plot for the west exhibits numerous peaks, reflecting significant variability in CWRULE distribution among provinces, which could be attributed to the geographic and climatic diversity of the region, as well as inconsistencies in responses to water resources management policies. Contour lines in the Northeast are closely packed and focused, indicating a high consistency in CWRULE, likely due to uniform and effective regional water resources management strategies.

### 3.4. Spatial convergence analysis of CWRULE

The spatial convergence analysis of CWRULE across the national level and four major regions, including both absolute and conditional β convergence, is presented in Table 3.

From the absolute β convergence (Panel A), the national β value of -0.248 indicates significant absolute convergence across the country, with a convergence speed of 2.85% and a half-life of 24.319 years. The strength of convergence varies across regions: the East shows weak convergence with a β of -0.039, a convergence speed of only 0.398%, and a prolonged

Table 3. Spatial convergence results of CWRULE.

| Variable | Panel A: Absolute β Convergence | | | | |
|---|---|---|---|---|---|
| | National | The four major regions | | | |
| | | East | Central | West | Northeast |
| β | -0.248*** | -0.039** | -0.451** | -0.396*** | -0.441*** |
| | (0.042) | (0.050) | (0.098) | (0.085) | (0.281) |
| Fixed effect | Yes | Yes | Yes | Yes | Yes |
| $R^2$ | 0.405 | 0.228 | 0.352 | 0.467 | 0.551 |
| Convergence speed (%) | 2.85 | 0.398 | 5.997 | 5.042 | 7.248 |
| Half-life convergence period | 24.319 | 174.241 | 11.559 | 13.748 | 9.564 |
| Variable | Panel B: Conditional β Convergence | | | | |
| | National | The four major regions | | | |
| | | East | Central | West | Northeast |
| β | -0.298*** | -0.026*** | -0.774*** | -0.751*** | -0.614** |
| | (0.041) | (0.067) | (0.106) | (0.070) | (0.023) |
| Control variables | Yes | Yes | Yes | Yes | Yes |
| Fixed effect | Yes | Yes | Yes | Yes | Yes |
| $R^2$ | 0.217 | 0.083 | 0.253 | 0.253 | 0.954 |
| Convergence speed (%) | 3.538 | 0.263 | 14.872 | 13.903 | 11.899 |
| Half-life convergence period | 19.590 | 263.114 | 4.661 | 4.986 | 5.825 |

Note: ***, **, and *respectively indicate significance levels above 1%, 5%, and 10%; standard deviations are shown in parentheses.

half-life of 174.241 years. This is consistent with the phenomenon of diminishing returns in developed regions, where high baseline efficiency of water resource utilization limits further improvement [40]. In contrast, the Central region exhibits strong convergence with a β of -0.451, a speed of 5.997%, and a half-life of 11.559 years, which may be attributed to increased investments in water management infrastructure and policy support. This trend is similar to that observed in other developing countries, where infrastructure development plays a crucial role in improving water management efficiency [41]. The West shows similar trends, with a β of -0.396, a speed of 5.042%, and a half-life of 13.748 years, aligning with findings that suggest increased funding and policy interventions can significantly enhance water resource utilization efficiency in underdeveloped areas [42]. Despite lower statistical significance, the Northeast region shows a high convergence speed of 7.248%, with a β of -0.441 and a half-life of 9.564 years. This may result from recent efforts to modernize industrial water management systems, a trend observed in other industrial regions undergoing structural adjustments [43].

From the conditional β convergence (Panel B), the national β coefficient improves to -0.298 after introducing control variables, indicating stronger conditional convergence with a convergence speed of 3.538% and a half-life of 19.59 years. This result is consistent with studies that emphasize the critical role of control variables such as economic development, technological progress, and policy support in enhancing water resource management efficiency [44]. At the regional level, the East exhibits very weak conditional convergence with a β of -0.026, a speed of 0.263%, and a half-life of 263.114 years, suggesting that even with control variables, significant barriers to improving water resource management efficiency persist in this region. This finding is in line with the challenges faced by developed regions, where complex administrative structures and high baseline efficiency limit the effectiveness of additional interventions [45].

In contrast, the Central region demonstrates strong conditional convergence, with a β of -0.774, a speed of 14.872%, and a half-life of 4.661 years. This rapid convergence may

be due to effective policy implementation and economic growth, which have been shown to drive improvements in water resource management efficiency in similar contexts [46]. The West also shows significant conditional convergence, with a β of -0.751, a speed of 13.903%, and a half-life of 4.986 years, highlighting the impact of regional development strategies, such as technological support and infrastructure investment, in enhancing water resource management efficiency [47]. Similarly, the Northeast records a β of -0.614, a convergence speed of 11.899%, and a half-life of 5.825 years, indicating substantial progress in improving water resource efficiency through the modernization of water management practices, consistent with trends observed in other industrial regions undergoing transformation.

The Eastern region is particularly noteworthy in terms of CWRULE convergence. From the results of absolute β convergence (Panel A), the convergence speed of CWRULE is only 0.398%, indicating significantly weak convergence. This suggests that, despite the relatively high baseline efficiency of water resource utilization in the Eastern region, further improvement is constrained by the "law of diminishing returns," where higher baseline efficiency limits additional progress. From the results of conditional β convergence (Panel B), after controlling for relevant variables such as economic development, technological progress, and policy support, the convergence speed of CWRULE in the Eastern region remains as low as 0.263%, with significant barriers still hindering improvements in water resource management efficiency. These challenges are closely tied to the region's socioeconomic and policy factors. First, the Eastern region, being highly developed with advanced industrialization and urbanization, is dominated by high water-consuming industries that create sustained high water demand. However, efficiency improvements in such industries often require significant technological investments, which are difficult to achieve in the short term. Second, the region's complex administrative structures and coordination challenges among multiple stakeholders weaken the effectiveness of policy implementation. Furthermore, existing policies in the Eastern region tend to focus more on maintaining current high efficiency rather than driving innovation for breakthroughs. To address these challenges, the Eastern region must prioritize innovative strategies to enhance CWRULE. This includes the development and adoption of advanced water management technologies, such as intelligent water systems, real-time monitoring tools, and big data analytics, to achieve digitalized and precise resource management. Additionally, stricter water-saving standards should be enforced alongside incentive-based policies, such as tax reductions and water rights trading, to encourage technological upgrades in water-intensive industries. Strengthening cross-regional coordination is also essential to optimize resource allocation through water redistribution, while expanding water rights trading pilots can further explore market-based mechanisms to improve CWRULE. These targeted strategies will help the Eastern region overcome its "efficiency improvement bottleneck" and achieve sustainable optimization in water resource management.

These results indicate that the inclusion of control variables significantly enhances the convergence speed and significance in the Central, West, and Northeast regions, reflecting the positive impact of factors such as policy support, economic development, and technological advancement on improving CWRULE. In contrast, the weak convergence observed in the East region in both absolute and conditional terms suggests the need for more targeted strategies to improve water resources management and utilization efficiency in this area. Furthermore, variations in R² values demonstrate the model's capacity to explain CWRULE variability across regions, with the Northeast showing the strongest explanatory power in conditional β convergence, indicating that changes in CWRULE in this region are more influenced by the included control variables.

## 4. Conclusions, policy implications and research prospects

Based on the analyses conducted, this paper arrives at the following conclusions:

(1) CWRULE measurements reveal significant and persistent disparities in water resource management across China, highlighting the urgent need to address regional imbalances. Central provinces such as Shanxi and Jiangxi demonstrate consistent improvements in water utilization efficiency due to infrastructure investments and policy support. In stark contrast, western regions like Gansu and Ningxia continue to lag significantly behind, with low efficiency levels and inadequate management practices, underscoring the need for immediate and focused interventions. Coastal provinces, including Beijing and Shanghai, benefit from advanced management systems and high levels of investment, resulting in significantly higher CWRULE values. However, such advanced practices have not yet been effectively transferred to less developed regions, exacerbating existing inequalities. (2) Analysis through kernel density estimation indicates a national trend of annual improvements in water resource utilization efficiency, particularly after 2016, reflecting the impact of advancements in policies, technologies, and management systems. A clear shift from lower to higher CWRULE values is observed in central and northeastern regions, driven by coordinated policy measures and economic development. However, the continued concentration of lower efficiency values in western regions suggests a pressing need for targeted policy adjustments and technological support to avoid further widening of the gap. This disparity highlights the risk of underperforming regions falling further behind if immediate action is not taken. (3) Convergence analysis reveals increased speed and significance in central, western, and northeastern regions when controlling for variables such as policy support, economic development, and technology adoption, showcasing the success of these factors in driving regional improvements. Nevertheless, the eastern region exhibits weak convergence despite its high baseline efficiency, which can be attributed to diminishing returns and structural barriers. This calls for innovative and adaptive strategies tailored to the region's specific challenges, including addressing complex administrative structures and fostering technological breakthroughs.

In light of these findings, it is critical to emphasize the urgency of addressing the pronounced disparities in water resource management. Underperforming regions, particularly in western China, require rapid, targeted, and region-specific interventions. This includes accelerated investment in infrastructure, technology transfer from developed regions, and the strengthening of local governance capacity. Without such actions, the inefficiencies in water utilization in these regions will not only hinder their economic and social development but may also compromise the overall sustainability of China's water resource management system. Bridging these gaps through tailored interventions is imperative to ensure equitable and sustainable utilization of water resources across the nation.

Based on the conclusions drawn from this research, the following policy recommendations are proposed: (1) Strengthening regional water resource management requires increased investment in infrastructure and technological upgrades, particularly in underperforming areas such as the western and northeastern regions, while also providing targeted policy support for the eastern region. In arid areas like the western region, priority should be given to groundwater recharge and reservoir construction projects, drawing on California's drought management experience, where Managed Aquifer Recharge (MAR) programs and integrated surface and groundwater management have effectively alleviated water scarcity. In water-rich areas such as the central and northeastern regions, advanced irrigation techniques like drip and sprinkler irrigation should be promoted, inspired by Israel's innovative agricultural water management practices that enhance efficiency in agricultural water use. For the eastern region, where water resource utilization efficiency is already high and constrained by the "law of diminishing marginal returns," the focus should be on adopting

advanced water management systems, such as smart water meters, real-time monitoring tools, and data analytics platforms, to optimize the utilization of existing resources. Additionally, policies should encourage the eastern region to strengthen water reuse and recycling in water-intensive industrial sectors. Singapore's "NEWater" initiative, which recycles industrial wastewater into high-quality reclaimed water, serves as a valuable model for improving water reuse efficiency in high-consumption industries.(2) Promote the adoption of advanced water management technologies by integrating smart water systems, real-time monitoring tools, and data analytics platforms into local water management frameworks. For example, smart metering systems, as implemented in California, enable real-time monitoring of water usage, helping identify inefficiencies and optimize distribution. These technologies could significantly enhance CWRULE, particularly in regions where water losses are high due to outdated infrastructure. In rural and underdeveloped areas, partnerships with domestic and international institutions can facilitate technology transfer and training programs to ensure sustainable outcomes. Tailored implementation of these technologies can help address specific regional challenges and improve overall water management efficiency. (3) Strengthen the enforcement of water resource policies by establishing specialized regulatory bodies to monitor compliance and penalize violations. Drawing inspiration from Australia's Murray-Darling Basin Authority, China could establish centralized water management frameworks to track water rights, withdrawals, and allocations, ensuring sustainable usage across regions. California's strict monitoring and enforcement measures during droughts, including financial penalties for overuse, also serve as a model for enhancing compliance. Transparent reporting mechanisms should be established to assess the effectiveness of water policies regularly, while adaptive governance frameworks would ensure timely revisions to address emerging challenges. Strengthening enforcement mechanisms would significantly contribute to improving CWRULE in regions with low water resource efficiency. (4) Increase public awareness and participation by launching nationwide educational campaigns on water conservation and engaging local communities in water management activities. Campaigns like California's "Save Our Water" initiative during drought periods effectively raised public awareness and reduced water use. China could adopt similar campaigns to promote household and industrial water efficiency, particularly in regions with low CWRULE values. Workshops on rainwater harvesting, greywater reuse, and efficient water use practices can help disseminate knowledge. Additionally, community-led initiatives, such as watershed management programs observed in India, can empower local stakeholders to take active roles in improving regional water resource management and foster a shared responsibility for water conservation. (5) Ensure continuous monitoring and research by establishing a comprehensive national water resource data platform. This platform should integrate data on water availability, usage, and quality to support research, policy-making, and public access. For example, the European Environment Agency's water information system demonstrates how accessible data can improve resource management decisions. Encouraging interdisciplinary research on emerging challenges, such as climate change impacts, would further guide policy development. California's collaboration with academic institutions on climate modeling to improve drought management provides a relevant example for enhancing CWRULE across China through science-based decision-making. By integrating successful strategies from regions such as California, Israel, Singapore, and Australia, these recommendations address the disparities in CWRULE across China. They aim to promote equitable and sustainable water resource utilization, ensuring improved efficiency in underperforming regions while maintaining progress in high-performing areas.

This research provides valuable insights into CWRULE in China, particularly highlighting regional disparities and the effectiveness of various management strategies. Comparisons with similar studies in other countries reveal important patterns; for instance, research in India

has shown significant regional differences in water resource utilization due to variations in infrastructure and policy implementation, emphasizing the need for localized strategies [38]. Similarly, a study in Brazil indicated that effective governance and technological advancement can markedly improve water management efficiency in certain regions [39]. These findings underscore the universality of the challenges faced in water resource management and the importance of tailored approaches across different contexts.

However, this study also has its limitations. A significant constraint is the reliance on secondary data, which may vary in accuracy due to differences in reporting standards among provinces. Additionally, while the temporal scope of the analysis is comprehensive, it does not capture long-term trends that could provide deeper insights into changes over time. Furthermore, the focus on quantitative metrics potentially overlooks qualitative factors, such as community engagement and cultural attitudes toward water conservation.

To address these gaps, future research should explore several areas: conducting longitudinal studies to understand how CWRULE evolves and the long-term effects of policies; incorporating qualitative methods, such as interviews and case studies, to capture community perspectives and identify local challenges; expanding the scope to include cross-national comparisons that highlight best practices in water resource management; and assessing the implications of climate change on water availability and management strategies. By addressing these areas, future studies can fill existing gaps and enhance understanding of effective water resource management strategies, ultimately contributing to more sustainable practices globally.

## Author contributions

**Conceptualization:** Xiongtian Shi, Chao Li.

**Data curation:** Xiongtian Shi.

**Formal analysis:** Xiongtian Shi.

**Funding acquisition:** Xiongtian Shi.

**Investigation:** Xiongtian Shi, Chao Li.

**Methodology:** Xiongtian Shi.

**Project administration:** Xiongtian Shi, Zhengyong Yu.

**Resources:** Xiongtian Shi, Zhengyong Yu.

**Software:** Chao Li.

**Supervision:** Chao Li.

**Validation:** Xiongtian Shi, Zhengyong Yu.

**Visualization:** Xiongtian Shi.

**Writing – original draft:** Xiongtian Shi, Zhengyong Yu.

**Writing – review & editing:** Xiongtian Shi.

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
