## [Decision Letter · Decision Letter 0]

19 Aug 2024

PONE-D-24-27008Evaluating the Comprehensive Water Resources Utilization Level in China: Dynamic Distribution Analysis and Spatial Convergence InsightsPLOS ONE

Dear Dr. shi,

Thank you for submitting your manuscript to PLOS ONE. After careful consideration, we feel that it has merit but does not fully meet PLOS ONE’s publication criteria as it currently stands. Therefore, we invite you to submit a revised version of the manuscript that addresses the points raised during the review process.

**ACADEMIC EDITOR: ** **Decision:** Major Revision Please find the reviewer comments below.

We look forward to receiving your revised manuscript.

Kind regards,

Manash Jyoti Bhuyan, M.A., M.Phil., Ph.D.

Academic Editor

PLOS ONE

 [The Yunnan University Research Foundation Program(Grant No. KC-23233830); Youth Project of Humanities and Social Science Foundation of Ministry of Education (22YJC790039).].  

3. We note that your Data Availability Statement is currently as follows: [All relevant data are within the manuscript and its Supporting Information files]

Additional Editor Comments:

The reviewers acknowledge the manuscript's significance but recommend major revisions to enhance its quality and clarity. Reviewer 1 suggests improving the abstract with an introductory statement, quantitative results, and a conclusion, addressing formatting inconsistencies, enriching the introduction with recent literature and clear research objectives, and reorganizing the methodology into distinct sections. The results need higher-resolution maps, revised graph titles, and a comprehensive discussion and conclusion.

Reviewer 2 emphasizes the need for formatting consistency, specific quantitative results in the abstract, and a detailed explanation of data sources and methodology. Additional improvements include eliminating repetitive content, ensuring consistent terminology, expanding the discussion to compare with similar studies, and addressing policy implications and future research directions.

Along with addressing these comments, ensure that the manuscript adheres to the journal's style and format, with properly titled tables and figures and accurately cited sources.

Reviewers' comments:

Reviewer's Responses to Questions

**Comments to the Author**

1. Is the manuscript technically sound, and do the data support the conclusions?

Reviewer #1: Yes

Reviewer #2: Yes

2. Has the statistical analysis been performed appropriately and rigorously? 

Reviewer #1: Yes

Reviewer #2: Yes

3. Have the authors made all data underlying the findings in their manuscript fully available?

Reviewer #1: Yes

Reviewer #2: Yes

4. Is the manuscript presented in an intelligible fashion and written in standard English?

Reviewer #1: Yes

Reviewer #2: Yes

5. Review Comments to the Author

Reviewer #1: Abstract:

• The abstract lacks an introductory statement and conclusion. Begin the abstract with a brief introduction to the topic, followed by the main objectives, key findings, and a concluding sentence summarizing the implications of the research.

• Keywords are too long. Use shorter and more specific keywords that are directly relevant to the study.

Introduction:

• The introduction section needs to incorporate recent empirical literature from around the world related to the topic. Review and include recent studies that discuss water resource utilization and spatial convergence to provide a comprehensive background.

• Research gaps are not clearly identified. Clearly articulate the existing research gaps that this study aims to address.

• Objectives are missing from the end of the introduction. List the specific objectives of the study at the end of the introduction section to provide a clear roadmap for the reader.

Methodology:

• The methodology section is good but lacks organization (i.e., Data collection, processing, and analysis). Clearly separate the methodology into subsections such as Data Collection, Data Processing, and Data Analysis for better organization.

• Mixing of methodological flow chart with the conceptual framework. Create a separate methodological flow chart to visualize the steps taken in the study, and keep the conceptual framework distinct.

Results:

• All maps need to be regenerated with a higher DPI of 1000 resolution to make the maps and legends clear. Ensure all maps are of high resolution to improve readability and clarity.

• Titles for each line graph on Figure 4 are not needed. Use (a), (b), etc., on the graphs and provide a proper caption for each letter, avoiding individual titles for clarity.

Discussion: The discussion section is missing from the manuscript. Add a discussion section that interprets the results, compares them with existing literature, and discusses the implications and limitations of the findings.

Conclusion: The conclusion is unclear and does not summarize the major findings. Write a clear conclusion that summarizes the major findings of the study in relation to the research questions or hypotheses. Highlight the implications and possible future research directions.

Overall Recommendation:

The manuscript is scientifically sound and addresses an important issue. However, significant revisions are needed to improve clarity, organization, and completeness. Incorporating these changes will enhance the manuscript's quality and readability, making it suitable for publication.

Reviewer #2: I have read the manuscript titled "Evaluating the Comprehensive Water Resources Utilization Level in China: Dynamic Distribution Analysis and Spatial Convergence Insights". The manuscript can be accepted for publication only after the authors agree to include major revisions. 

I have read the manuscript titled "Evaluating the Comprehensive Water Resources Utilization Level in China: Dynamic Distribution Analysis and Spatial Convergence Insights". The manuscript can be accepted for publication only after the authors agree to include major revisions. Here are some comments and observations on the manuscript: 1. Ensure the authors' affiliations are correctly formatted2. The abstract is comprehensive but could benefit from specific quantitative results or metrics to provide more concrete insights into the findings.3. The document shows inconsistencies in the formatting of titles, section headings, and paragraph spacing. For instance, "2．Research design" and "1.Introduction" display inconsistencies in numbering and punctuation.4. The introduction section can be further strengthened by providing more background information on the existing challenges in water resource management globally to set a broader context.5. Certain phrases and concepts are repeated unnecessarily. For example, the abstract and introduction both contain repetitive details regarding regional disparities and policy impacts.6. Ensure the consistent use of terms and acronyms throughout the manuscript, such as "CWRULE," to maintain clarity for the reader.7. Elaborate on the sources of the data used in the study and discuss any potential limitations or biases in the data. This will provide context and strengthen the validity of the findings.8. Include more details on the methodological framework, especially the rationale for choosing the Dagum Gini coefficient, spatial kernel density estimation, and spatial convergence models. Explain why these methods are suitable for the study.9. Ensure that all figures and tables are adequately labelled, captioned, and referenced in the text. Table name and number should be in the same line. Don’t start a sentence by writing “Fig.4 displays changes in the…….” Or end the sentence with “……..the calculated results are presented in Fig.6”. Instead cite the table or the figure within the text after a sentence by using brackets.10. Expand the discussion to include comparisons with similar studies or contrasting results. Discuss the limitations of the study and propose how future research can address these gaps.11. Discuss the implications of the findings for policymakers, especially concerning the identified regional disparities. Suggest specific policy measures that could address the issues highlighted by the study.12. Strengthen the conclusion by summarizing the key findings and emphasizing their importance. Also, suggest areas for future research that could build on the current study's findings. Implementing these corrections and suggestions will likely enhance the readability and impact of the manuscript, making it more accessible and informative to the audience. 

6. PLOS authors have the option to publish the peer review history of their article (what does this mean?). If published, this will include your full peer review and any attached files.

Reviewer #1: No

Reviewer #2: No

---

## [Author Response · Author response to Decision Letter 1]

10 Oct 2024

We are pleased to submit the revised version of our manuscript titled “Evaluating the Comprehensive Water Resources Utilization Level in China: Dynamic Distribution Analysis and Spatial Convergence Insights”for consideration of publication in PLOEONE.

We would like to express our sincere gratitude to the reviewers and the editorial team for their valuable comments and suggestions, which have greatly contributed to the improvement of our manuscript. In response to the reviewers' feedback, we have made the following significant revisions:

1.Abstract:

I have revised the abstract to include an introductory statement, providing a brief introduction to the topic. A concluding sentence has also been added to summarize the key findings and implications of the research.

2.Keywords:

The keywords have been revised to be shorter and more specific, ensuring they directly reflect the core aspects of the study.

3.Introduction:

Empirical Literature: Recent empirical studies on water resource utilization and spatial convergence from around the world have been incorporated to provide a comprehensive background and context.

Research Gaps: I have clearly articulated the research gaps that this study aims to address, outlining the existing limitations in the literature.

Research Objectives: Specific research objectives have been added at the end of the introduction to provide a clear roadmap for the readers.

4.Methodology:

Organization: The methodology section has been reorganized into subsections: Data Collection, Data Processing, and Data Analysis, for better clarity and structure.

Methodological Flow Chart: A separate methodological flow chart has been created to clearly visualize the research steps. The conceptual framework and the methodological flow chart have been distinguished to avoid confusion.

5.Results:

Map Resolution: All maps have been regenerated with a higher DPI of 1000 to ensure that the maps and legends are clear and readable.

Figure 4 Titles: The titles for each line graph in Figure 4 have been removed and replaced with (a), (b), etc., with a corresponding caption for each letter to improve clarity and avoid redundancy.

6.Discussion:

A discussion section has been added to interpret the results in relation to existing literature, and to discuss the implications and limitations of the findings.

7.Conclusion:

The conclusion has been rewritten to clearly summarize the major findings of the study in relation to the research questions. Implications of the research and potential future research directions have also been highlighted.

8.Overall Recommendation:

I sincerely appreciate the reviewers' constructive feedback. These revisions have significantly improved the clarity, organization, and completeness of the manuscript, enhancing its quality and readability.

The conclusion has been rewritten to clearly summarize the major findings of the study in relation to the research questions, and to highlight the implications and potential future research directions.

We believe that these revisions have significantly improved the quality and clarity of the manuscript. We hope that the revised version meets the expectations of the reviewers and the editorial board.

Thank you for considering our revised manuscript for publication in [Journal Name]. We look forward to your favorable response. Please feel free to contact us if you need any further information.

---

## [Decision Letter · Decision Letter 1]

23 Dec 2024

PONE-D-24-27008R1Evaluating the Comprehensive Water Resources Utilization Level in China: Dynamic Distribution Analysis and Spatial Convergence InsightsPLOS ONE

Dear Dr. shi,

Thank you for submitting your manuscript to PLOS ONE. After careful consideration, we feel that it has merit but does not fully meet PLOS ONE’s publication criteria as it currently stands. Therefore, we invite you to submit a revised version of the manuscript that addresses the points raised during the review process.

We look forward to receiving your revised manuscript.

Kind regards,

Manash Jyoti Bhuyan, M.A., M.Phil., Ph.D.

Academic Editor

PLOS ONE

Journal Requirements:

Additional Editor Comments:

Please revise the entire manuscript thoroughly, addressing the previous reviewer comments comprehensively. Ensure that all feedback is carefully considered and effectively incorporated. Importantly, it has come to my attention that parts of the manuscript include AI-generated content. If AI tools were used in drafting or editing, this must be explicitly acknowledged in the manuscript. Additionally, ensure that no AI-generated text has been copied and pasted directly without proper cross-checking for accuracy, originality, and relevance. For instance, generic phrases such as “Thank you for considering our revised manuscript for publication in [Journal Name]” appear to reflect standard AI-generated language and should be critically evaluated or rewritten to maintain authenticity and alignment with the manuscript's style. Please prioritize originality and accuracy throughout the revision process.

Reviewers' comments:

Reviewer's Responses to Questions

**Comments to the Author**

1. If the authors have adequately addressed your comments raised in a previous round of review and you feel that this manuscript is now acceptable for publication, you may indicate that here to bypass the “Comments to the Author” section, enter your conflict of interest statement in the “Confidential to Editor” section, and submit your "Accept" recommendation.

Reviewer #2: (No Response)

2. Is the manuscript technically sound, and do the data support the conclusions?

Reviewer #2: (No Response)

3. Has the statistical analysis been performed appropriately and rigorously? 

Reviewer #2: (No Response)

4. Have the authors made all data underlying the findings in their manuscript fully available?

Reviewer #2: (No Response)

5. Is the manuscript presented in an intelligible fashion and written in standard English?

Reviewer #2: (No Response)

6. Review Comments to the Author

Reviewer #2: The authors have addressed several critical revisions based on reviewer feedback, including reworking the abstract, updating keywords, expanding the literature review, reorganizing the methodology, and enhancing the discussion and conclusion sections. Although the paper needs some minor revisions to be incorporated:

1. The convergence analysis results could benefit from a more detailed discussion on why specific regions like the East show weaker convergence. Exploring potential socio-economic or policy-based factors underlying these differences could lead to stronger regional policy recommendations.

2. The policy suggestions are well-developed but could include examples of successful policy models from other regions or countries that address similar issues (e.g., drought management in California). This would provide actionable insights beyond theoretical recommendations.

3. While the conclusion summarizes findings well, it could emphasize the urgency of water management disparities and the need for rapid intervention in underperforming regions to better highlight the study's importance.

7. PLOS authors have the option to publish the peer review history of their article (what does this mean?). If published, this will include your full peer review and any attached files.

Reviewer #2: No

---

## [Author Response · Author response to Decision Letter 2]

7 Jan 2025

We sincerely thank the reviewer for their valuable comments. We have made the following revisions:

 The authors have addressed several critical revisions based on reviewer feedback, including reworking the abstract, updating keywords, expanding the literature review, reorganizing the methodology, and enhancing the discussion and conclusion sections. Although the paper needs some minor revisions to be incorporated:

1.The convergence analysis results could benefit from a more detailed discussion on why specific regions like the East show weaker convergence. Exploring potential socio-economic or policy-based factors underlying these differences could lead to stronger regional policy recommendations.

We sincerely thank the reviewer for their valuable comments. In response to the issue of weaker convergence in the eastern region, we have conducted a more detailed discussion on the underlying socio-economic and policy factors. Due to the already high baselinethe comprehensive water resources utilization level (CWRULE) in the eastern region, the "law of diminishing marginal returns" applies, meaning that further improvements in CWRULE are limited. Additionally, the highly developed industrialization and urbanization in the eastern region have led to a dominance of water-intensive industries, resulting in continuously high water demand. However, improvements through high-tech investments are often difficult to achieve in the short term. Furthermore, the complex administrative structures and coordination challenges among multiple stakeholders have weakened the effectiveness of policy implementation. Existing policies tend to focus on maintaining high CWRULE rather than supporting innovative breakthroughs. To address this, we have added specific policy recommendations for the eastern region, including the development and promotion of innovative water management technologies such as smart water systems and real-time monitoring tools, the implementation of stricter industry water-saving policies (e.g., tax incentives and water rights trading), and the strengthening of cross-regional coordination and resource optimization. These analyses and recommendations further deepen our understanding of the challenges in the eastern region and provide practical solutions for improving its CWRULE.

2. The policy suggestions are well-developed but could include examples of successful policy models from other regions or countries that address similar issues (e.g., drought management in California). This would provide actionable insights beyond theoretical recommendations.

We sincerely thank the reviewer for their valuable comments. In response to the reviewer’s suggestion to “incorporate successful cases from other regions or countries,” we have added several international policy models and their insights to the policy recommendation section. For example:

(1)California’s drought management: We highlighted its Managed Aquifer Recharge (MAR) programs, which effectively alleviated water shortages during droughts by restoring groundwater reserves. Additionally, California’s “Save Our Water” campaign significantly raised public awareness of water conservation, playing a crucial role in drought management. These experiences offer valuable lessons for addressing water scarcity in China’s western regions.

(2)Israel’s agricultural irrigation technology: Israel has significantly improved agricultural water efficiency through advanced irrigation systems, such as drip and sprinkler irrigation. This provides a useful reference for China’s central and northeastern regions, which are water-rich but have lower efficiency in agricultural water use.

(3)Singapore’s “NEWater” initiative: This program achieves the recycling of wastewater into industrial and potable water, offering inspiration for water-saving policies in China’s water-intensive industrial sectors.

(4)Australia’s basin management experience: The Murray-Darling Basin Authority’s cross-regional coordination mechanism ensures the sustainable allocation of water resources, providing practical insights for China to establish cross-regional water rights management and resource allocation policies.

The incorporation of these international success stories not only enhances the practical applicability of the policy recommendations but also provides diverse solutions to address China’s regional water resource management challenges.

3. While the conclusion summarizes findings well, it could emphasize the urgency of water management disparities and the need for rapid intervention in underperforming regions to better highlight the study's importance.

We sincerely thank the reviewer for their valuable comments. In the conclusion section, based on the reviewer’s suggestion to "emphasize the urgency of addressing water resource management imbalances and the necessity of rapid intervention," we have highlighted the significant disparities in water resource utilization efficiency (CWRULE) across different regions in China. Specifically, the low water resource management efficiency in the western and northeastern regions poses potential threats to regional development and the sustainable utilization of water resources nationwide. The paper further proposes that prompt intervention is needed in underperforming regions, including increased investment in infrastructure, promotion of efficient water-use technologies, and optimization of policy support, to narrow the regional gaps. Additionally, the study emphasizes its significance by not only identifying the disparities in regional water resource management but also providing theoretical foundations for formulating region-specific water resource policies. These revisions make the study’s conclusions more aligned with real-world challenges and underscore its value in guiding policymaking.

By revising the sections on convergence analysis, policy recommendations, and conclusions, the paper further deepens its analysis of the research problem, incorporates international perspectives, and strengthens the practical applicability of the policy recommendations. These adjustments enhance the importance and urgency of the research, making the paper more rigorous and solution-oriented. We sincerely thank the reviewer for their valuable comments, which have greatly contributed to the improvement of our manuscript.

2024.12.27

---

## [Editor Report · Decision Letter 2]

15 Jan 2025

PONE-D-24-27008R2Evaluating the Comprehensive Water Resources Utilization Level in China: Dynamic Distribution Analysis and Spatial Convergence InsightsPLOS ONE

Dear Dr. shi,

Thank you for submitting your manuscript to *PLOS ONE*. After careful evaluation, we have determined that it requires **Minor Revisions**, which may take 1-2 days to complete. Therefore, we invite you to submit a revised version of the manuscript that addresses the points raised during the review process.

We look forward to receiving your revised manuscript.

Kind regards,

Manash Jyoti Bhuyan, M.A., M.Phil., Ph.D.

Academic Editor

PLOS ONE

Journal Requirements:

Additional Editor Comments:

The manuscript is in an almost in a favourable position for acceptance, provided the following revisions and modifications are addressed:

1. Remove generic sentences like, "The changes in the Gini coefficient for CWRULE and its decomposition are presented in Fig. 3." Instead, cite figures and tables within sentences that describe their content. That is, incorporate the figures and tables seamlessly into the narrative where the figure or the data from the Table is discussed. This approach should be applied consistently throughout the manuscript. You can refer to the following manuscripts which might help you in this regard. https://doi.org/10.1016/j.scitotenv.2023.167525, https://doi.org/10.1016/j.envsci.2024.103862, https://doi.org/10.57372/00009752

2. The Introduction section would benefit from a more detailed discussion and appropriate citations on global water utilization and management practices. Along with the references already mentioned, the following paper would be particularly useful in this context: https://doi.org/10.1016/j.jhydrol.2024.131328. Additionally, more citations are required to provide a comprehensive overview and to strengthen the arguments presented.

3. The sentence "The minimal dataset for this study has been uploaded as a supplementary file..." should be revised and moved to a separate section under the heading "Data Availability" for better clarity and alignment with standard manuscript organization.

---

## [Author Response · Author response to Decision Letter 3]

27 Jan 2025

1. Remove generic sentences like, "The changes in the Gini coefficient for CWRULE and its decomposition are presented in Fig. 3." Instead, cite figures and tables within sentences that describe their content. That is, incorporate the figures and tables seamlessly into the narrative where the figure or the data from the Table is discussed. This approach should be applied consistently throughout the manuscript. You can refer to the following manuscripts which might help you in this regard. https://doi.org/10.1016/j.scitotenv.2023.167525, https://doi.org/10.1016/j.envsci.2024.103862, https://doi.org/10.57372/00009752

Thank you for your valuable suggestion. We fully understand the importance of removing generic statements such as "Fig. 3 presents the changes in the Gini coefficient for CWRULE and its decomposition" and instead incorporating figure references more naturally into the narrative when describing data or discussions. This approach can enhance the narrative coherence and readability of the manuscript. Following the style of the references you provided, we have revised the manuscript accordingly, integrating figure references into specific data analyses and discussions rather than describing the figures separately.

2.The Introduction section would benefit from a more detailed discussion and appropriate citations on global water utilization and management practices. Along with the references already https://doi.org/10.1016/j.jhydrol.2024.131328. Additionally, more citations are required to provide a comprehensive overview and to strengthen the arguments presented.

Thank you for your valuable suggestion. We fully agree that adding a more detailed discussion on global water utilization and management practices, along with appropriate citations, would enhance the comprehensiveness and persuasiveness of the Introduction section. We have reviewed the paper you recommended (https://doi.org/10.1016/j.jhydrol.2024.131328) and incorporated it into the revised Introduction. Additionally, we have included other relevant references to provide a more thorough background and supporting evidence. The revised manuscript now includes a more detailed discussion of global water management practices and strengthens the scientific basis of the arguments presented.

3. The sentence "The minimal dataset for this study has been uploaded as a supplementary file..." should be revised and moved to a separate section under the heading "Data Availability" for better clarity and alignment with standard manuscript organization.

Thank you for your valuable suggestion. We fully agree that moving the sentence "The minimal dataset for this study has been uploaded as a supplementary file..." to a separate "Data Availability" section would more clearly convey the data availability information and better align with the standard structure of a manuscript. We have made the corresponding revision as suggested.

---

## [Editor Report · Decision Letter 3]

29 Jan 2025

Evaluating the Comprehensive Water Resources Utilization Level in China: Dynamic Distribution Analysis and Spatial Convergence Insights

PONE-D-24-27008R3

Dear Dr. shi,

We’re pleased to inform you that your manuscript has been judged scientifically suitable for publication and will be formally accepted for publication once it meets all outstanding technical requirements.

Kind regards,

Manash Jyoti Bhuyan, M.A., M.Phil., Ph.D.

Academic Editor

PLOS ONE
---

## [Editor Report · Acceptance letter]

PONE-D-24-27008R3

PLOS ONE

Dear Dr. Shi,

I'm pleased to inform you that your manuscript has been deemed suitable for publication in PLOS ONE. Congratulations! Your manuscript is now being handed over to our production team.

Kind regards,

on behalf of

Dr. Manash Jyoti Bhuyan

Academic Editor

PLOS ONE